# Preliminary Investigation and Analysis of Beachgoers’ Awareness of Rip Currents in South China

**DOI:** 10.3390/ijerph20054471

**Published:** 2023-03-02

**Authors:** Daoheng Zhu, Zhiqiang Li, Pengpeng Hu, Bingfu Wang, Qianxin Su, Gaocong Li

**Affiliations:** 1School of Electronics and Information Engineering, Guangdong Ocean University, Zhanjiang 524088, China; 2School of Liberal Arts, Kookmin University, Seoul 02707, Republic of Korea

**Keywords:** coastal hazards, rip current, preliminary investigation, educational strategy

## Abstract

Among many coastal hazards, rip currents have gradually become one of the most noticeable hazards. Studies have demonstrated that most drowning accidents at beaches around the world are related to rip currents. In this study, online and field questionnaires were combined for the first time to reveal beachgoers’ awareness of rip currents in China from four aspects: demographic characteristics, swimming ability, information about visiting beaches, and knowledge about rip currents. One educational strategy was introduced to the field survey. The results suggest that (i) the proportion of online and field respondents who have heard of “rip currents” and seen warning signs of rip currents is extremely small. This reflects that beachgoers lack awareness of rip current hazards. Thus, China needs to strengthen the safety education of rip current knowledge. (ii) The level of awareness of rip currents can significantly affect the community’s ability to identify the location of rip currents and their choice of escape direction. (iii) In the field survey, we implemented an educational strategy as an intervention for respondents, and the accuracy of identifying rip currents and choosing the correct escape route improved by 34% and 46.7%, respectively. This implies that the intervention of educational strategy can significantly deepen beachgoers’ awareness of rip currents. It is recommended that more educational strategies about rip current knowledge be implemented on Chinese beaches in the future.

## 1. Introduction

Coastal tourism indicates the tourism operation and service activities based on various natural and cultural landscapes in coastal zones, islands, and oceans, ranking first in China’s ocean industry for a long time [1]. In 2020, coastal tourism accounted for 47.0% of the economic added value of the ocean industry in China. This is more than twice that of the second largest maritime economic industry (the marine transportation industry), though it decreased by 24.5% compared with the previous year due to COVID-19 [2]. Moreover, the rapid development of coastal tourism is accompanied by the frequent occurrence of some beach hazards. For example, rip currents (often called “rips” or “rip tides”) are fast, narrow, and strong seaward flows that arise from alongshore variations in wave setup, landward of the surf zone. Due to their dependence on wave breaking, rips can extend from close to the shoreline, through the surf zone, and varying distances beyond. They are ubiquitous on wave-exposed coasts in oceans, seas, and great lakes [3]. This is, despite rips, a fast way for sediments, plankton, nutrients and other materials to enter the sea; they are also a typical coastal hazard that can carry swimmers and surfers into deep water, which can lead to drowning accidents [4]. The rips can be robustly classified into three broad categories based on the dominant controlling forcing mechanism: hydrodynamically-controlled, bathymetrically-controlled, and boundary-controlled [3].

Rips and drowning accidents are frequently reported on many tourist beaches around the world [5,6]. Since the rip current hazard has long been threatening the life safety of beach users, it has brought about many negative effects on beach management and recreational activities [7]. For example, rips are a major coastal hazard in the US and Australia, and 60–80% of drowning accidents are directly associated with rips according to incomplete statistics [8,9]. The statistics of the ROYAL National Lifeboat Institution indicate that about 67% of rescues on UK beaches each year are caused by rips [10,11]. In India, the annual average number of deaths induced by rips from 2000 to 2010 was over 39 [12]. Researchers also conducted a survey on the beaches of southern Brazil and revealed that the hazards ascribed to rips account for 78% of the total annual beach hazards [13]. In Japan alone, more than 300 drowning accidents due to rips were reported between 2003 and 2011 [14]. South Korea is also one of the countries most severely affected by rips. Every summer, dozens of people are swept away by rips and subsequently require rescue, especially on the Haeundae Bathing Beach in Busan City in South Korea [15,16]. Moreover, more than 30 drowning accidents occurred at Pahang Beach in Malaysia from 2006 to 2018, owing to rips after investigation [17,18]. Therefore, rips have gradually attracted high attention from government authorities, surfers, and swimmers in many countries [4,19].

Currently, several beach surveys have been conducted in Australia, the United States, the United Kingdom, Costa Rica, Brazil, New Zealand, and Malaysia [10,20,21,22,23,24,25,26,27]. The survey content includes visitors’ demographics, swimming ability, frequency of visiting beaches, awareness of rips and warning signs, and whether they had been trapped in the rip current. The researchers presented images with and without rips and then asked visitors to point out the location of rips on the images or choose an area where they thought it was safe to swim. After evaluating beachgoers’ beliefs and behaviors about beach flags and rips, Sherker et al. [20] discovered that while those with a rudimentary understanding of rips were better able to recognize and avoid them, beachgoers with children were more likely to swim between flags. In accordance with the findings of Gallop et al.’s study [27], Caldwell et al. [21] showed that novice or uninformed beachgoers concluded that the calm water region was the safest swimming location to a greater extent. According to Matthews et al. [28], many beachgoers ignored the caution notices posted at the entrances to the beaches. By their analysis, Brannstrom et al. [22] were able to confirm that beach warning signs did not effectively educate beachgoers on important information. Woodward et al. [10] investigated beachgoers’ awareness of rip currents on four UK beaches, and proposed that only those who had personally experienced rips and received the necessary education could fully comprehend them, implying that it is crucial to inform beachgoers about rips and publicize this information. Similarly, the following groups were identified as high-risk by Fallon et al. [23]: young adults, foreign beachgoers, non-swimmers, and non-beachgoers. It was suggested that a campaign on rip current safety be directed at these groups in Miami Beach, while interventions outside of the home and educational institutions be introduced also. According to the research findings of Llopis et al. [24], 58% of respondents failed to see beach warning signs, and 40% were perplexed by the information on the signs. Education for people at high risk should be emphasized in addition to hiring professional lifeguard teams and creating better warning signs. Pitman et al. [25] objectively evaluated the respondents’ capacity to translate their propensity to spot rips in photographs into their propensity to spot them in real life. The results revealed that 66% of the respondents who could correctly identify the rips in the images still were unable to correctly identify them on site. In a survey of beachgoers at Cavendish and Barkely beaches in Canada, Locknick et al. [29] revealed that respondents’ impressions of rip currents might not accurately represent their actions, which are impacted by the layout of the beach access and other visitors. To help individuals better recognize rips in real scenes, 3D/VR equipment or videos of real rips should be incorporated into future education. For instance, a beach safety-related APP can assist in keeping beachgoers updated on the situation and potential risks. It can also be used as a platform for education in the meantime.

Specifically, 11 coastal provinces and cities in China cover 18, 000 km of coastline, with hundreds of natural beaches. Every year, a large number of beachgoers and residents are involved in accidents in the surf zone, resulting in either rescue or drowning. According to incomplete statistics, the cumulative number of drowning deaths in China’s surf zones from 2010 to 2019 reached 660, which even exceeded the number of casualties (628) caused by marine hazards in the same period [7]. Although China’s National Marine Hazard Mitigation Service (NMHMS) has launched a nationwide program to prevent rip current hazards, there has not been a survey of beachgoers’ awareness of rip current hazards. Therefore, the questionnaire and on-the-spot questionnaire investigation are employed in this research to survey the public. This is the first public survey of beachgoers at popular beaches in South China, combining an online survey with a field survey. This study aims to (i) understand beachgoers’ demographics, swimming ability, information on visiting beaches, and awareness of rips, (ii) publicize knowledge about rip current hazard to the public, and (iii) assess our educational strategy through the results of the questionnaire.

## 2. Material and Methods

Informed consent was obtained from all the study participants. We obtained parental consent for the minors before the study began. All procedures under this study were performed in accordance with the Declaration of Helsinki and relevant policies in China.

### 2.1. Research Area

In this study, six famous beaches in South China were selected for the field survey, as displayed in Figure 1. The visitor volume and hydrological conditions of the six beaches were obtained by referring to the data [30,31], as listed in Table 1. Sediment samples were collected during the field survey, with sampling locations in the intertidal area on each beach profile. The littoral grain size was represented by the averaged sediment median diameter $D_{50}$ from the granulometric analysis of sampling.

Silver beach and Longhaitian beach are flat and straight beaches with regular diurnal tide and irregular semidiurnal tide, respectively. The rest of the four beaches are headland beaches, among which Golden beach has irregular semidiurnal tide; Dameisha beach, Holiday beach, and Dadonghai beach have irregular diurnal tide.

Silver beach and Longhaitian beach cover a large flat and open area and have soft and fine sands and a beautiful seascape, with an annual beachgoer reception of about 2 million and 1.2 million. The superior geographical location and convenient traffic conditions make Dameisha beach one of the most famous coastal beachgoer attractions in southern China, with an annual beachgoer reception of about 4 million. Holiday beach is the largest tourist beach in Haikou City and is located in the tropical monsoon climate zone, with a pleasant climate all year round. Its annual visitor volume reaches 1.2 million. Dadonghai beach is one of the most famous recreational beaches in Sanya, and it receives more than 1.8 million visitors every year. These scenic spots are open for free all year round, with a huge number of beachgoers, especially on holidays. Thus, they are suitable as locations for questionnaire surveys.

### 2.2. Morphodynamic Beach Classification

The Australian beach classification model [32] essentially describes the morphodynamic response of the beach to the combined effect of wave, tide, and sedimentation. This morphodynamic analysis may be able to predict the likelihood of rips on natural beaches, but it is ineffective near estuaries or on beaches with artificial structures, where the wave and flow are dramatically altered [33]. The morphodynamic classification method is theoretically applicable because the six beaches chosen for this study do not have any man-made structures and are not close to estuaries.

The calculated dimensionless sediment fall velocity *Ω* and tide-wave parameter *RTR* divide the beach state into 8 types corresponding to different levels of rip current hazard [34,35], which are defined as Equations (1)–(4).
(1)$$\Omega =H_{b}/\left(T\omega_{s}\right)$$
(2)$$RTR=TR/H_{b}$$
(3)$$H_{b}=0.39g^{1/5}{\left(TH_{\infty }^{2}\right)}^{2/5}$$
(4)$$\omega_{s}=\left(RgD^{2}\right)/\left[C_{1}v+{\left(0.75C_{2}RgD^{3}\right)}^{0.5}\right]$$
where $H_{b}$ is the wave breaking height (m), $\omega_{s}$ represents the sediment fall velocity ($m\cdot s^{-1}$), *T* stands for the wave period (s), *TR* is the mean tide range (m), *g* is the gravity constant ($g=9.8\;m\cdot s^{-2}$), and $H_{\infty }$ is the deepwater wave height. *R* is the submerged specific gravity (1.65 for quartz in water), *D* represents the median diameter of the beach sediment, v is the kinematic viscosity of the fluid (($v=1.0*10^{-6}\mathrm{kg}\cdot m^{-1}\cdot s^{-1}$ for water at 20 °C), and $C_{1}$ and $C_{2}$ are constants. Ferguson and Church [36] recommend $C_{1}=18$ and $C_{2}=1$. The *Ω* indicates the mobility of the sediment which subsequently affects the nearshore morphology. A large value represents the dissipative beach state with high-level wave energy and fine sediments, while a small value represents the reflective beach with coarse sediment and small waves. The 𝑅𝑇𝑅 compares the contributions between wave and tide, separating the concave shoal under high-energy waves from the flat tidal terrace and the intermediate tidal-modified beach [7]. Table 2 lists the eight beach states with three rip current hazard levels corresponding to different parameters *RTR* and *Ω*.

### 2.3. Survey Methods

Our survey methods can be represented by the flow chart in Figure 2. We collected data through online and field questionnaires. Firstly, the questionnaire was written online based on four aspects: demographics, swimming ability, beach travel information, and awareness of rips. There were 15 questions in total, and the detailed contents were introduced in Appendix B.

The majority of the respondents for the online poll were chosen at random, and they completed it using either their smartphones or the Internet. The questionnaires were distributed through social media and published on the Internet beforehand using the site “Questionnaire Star” (https://www.wjx.cn/ (accessed on 8 July 2021)). When responders completed the questionnaire, information was gathered using the “Questionnaire Star” platform. We collected 1068 pieces of data during a one-month internet investigation. The field survey was conducted on the six beaches in Section 2. We randomly selected some visitors, and the purpose of our survey was explained by talking to visitors on the beach. If respondents agreed to fill in the questionnaire, then they were invited to answer questions 2 to 15 in Appendix B. For those respondents who had never heard of rips or could not understand the concept, a video (Appendix A) with rips was shown to the respondents, and the morphology and characteristics of rips were described according to the educational strategy in Section 2.4. After the intervention, those respondents were invited to answer questions 13 and 14 again (Appendix B). Finally, a total of 306 field survey data were obtained. Many respondents expressed that they had gained some knowledge and self-help skills, and provided recognition and encouragement for our work.

The data obtained from the online and field questionnaire were analyzed using a binary logistic model in SPSS. According to our model, the significance value of each independent variable with respect to the dependent variable was calculated, and p<0.05 was a statistically significant difference.

### 2.4. Educational Strategy

According to the results of the online questionnaire, the respondents who had not heard of rips and had not seen their warning signs accounted for a large proportion since they had not received relevant education or training. Therefore, the educational strategy was implemented for beachgoers in the field questionnaire survey. The survey process is described as follows. (1) Beachgoers’ demographics, swimming ability, information of visiting beaches, and awareness of rips were investigated following the questions listed in Appendix B. (2) The basic knowledge of rips was introduced to the respondents, and the video with rips was shown to them. A video was translated well in advance and annotated with Chinese subtitles. (3) Some examples were offered to illustrate the existence of rip current hazards on some beaches in China, and show that many drowning accidents have been caused. (4) The escape methods from rips were publicized: keep calm and swim parallel to the direction of the coastline, or drift with the current to a distance, then swim parallel to the direction of the coastline for a distance after the flow rate decreases, and finally swim back to the shore. (5) Respondents were invited to answer questions 13 and 14 again. Figure 3 exhibits the intervention work for beachgoers on the beach site.

### 2.5. Statistical Analyses

Two dependent variables and eleven independent variables were selected from the questionnaire, and the meaning and assignment of the respective variables are listed in Table 3. The respective independent variable received $\chi^{2}$ test with the dependent variable to determine its effect on the dependent variable. The independent variables impacting the dependent variable were covered in the model before the logistic regression analysis, while variables exerting no or a slight effect on the dependent variable were excluded from the model.

Since the dependent variables $Y_{1}$ and $Y_{2}$ were designed with only two final values of the results, complying with the binary logic, the binary logistic model could be adopted to test and estimate the degree of influence and significance level of the respective variable, etc. The empirical studies were conducted in models *M1* and *M2*, respectively. Furthermore, *M1* took correctly highlighting the location of rips ($Y_{1}$) as the dependent variable to examine the factors of respondents’ ability to identify rips in images. *M2* took escaping from the correct direction ($Y_{2}$) as the dependent variable to explore the factors of respondents’ awareness when they were trapped in rips. The model expressions are presented in Equation (5).
(5)$$ln\left(\frac{P_{i}}{1-P_{i}}\right)=\alpha +\overset{k}{\underset{k-1}{\sum }}\beta_{i}X_{ki}+\mu_{i}$$
where α denotes the intercept, $\beta_{i}$ is the regression coefficient, $X_{ki}$ is the kth explanatory variable, $P_{i}$ represents the probability of $Y_{i}$ corresponding to the final selection result of respondents, $1-P_{i}$ indicates the probability of non-occurrence, $\mu_{i}$ express the residual term. The variables with the value of *p* were less than 0.05, or those considered to be closely correlated with the dependent variable based on practical experience were selected with statistical significance by the results of the $\chi^{2}$ test. When the value of $P_{i}$ is less than 0.05, the model is considered to fit the data, and it is constructed effectively.

## 3. Results

### 3.1. Morphodynamic Classification Results

Using morphodynamic analysis, we calculated the beach states and occurrence probability of rip current hazards for the six beaches in Section 2. The beach states of Silver beach were ultra-dissipative, and Longhaitian beach was barred dissipative in most months. The beach states of Golden beach were barred dissipative and non-barred dissipative, and Dameisha beach was reflective on the whole. The beach states of Holiday beach were low tide terrace, and Dadonghai beach were low tide bar/rip and barred. The hazard levels of Longhaitian and Dadonghai beaches were rated as medium and medium to high, respectively, and Golden beach was rated as low to medium, and others were rated as low (see Figure 4). Moreover, our results, as judged by satellite images, were consistent with the results of the morphodynamic analysis (see Figure 5).

### 3.2. Survey Results

Regarding the origin of respondents, the number of respondents decreased from the eastern coastal areas to the western inland areas, covering most provinces in China. Therefore, these data are highly representative and extensive.

There were 552 males and 516 females, accounting for 51.7% and 48.3%, with an average age of 32.28 and 32.14, respectively. In terms of determining the location of rips, males aged 36–59 had the highest accuracy rate (65.1%), while those over 60 years old had the lowest accuracy rate (7.1%) in choosing the direction of escape. The results demonstrated that gender difference may not be a factor influencing the identification of rip location and the choice of escape direction.

Age groups were divided according to China’s national population segmentation standards: under 11 years for children, 12–18 for teenagers, 19–35 for youth, 36–59 for middle age, and 60 and above for the elderly. The number of young respondents aged 19–35 was the largest, accounting for 75.7% of the total. Concerning educational attainment, 65.7% of the total number of respondents had a bachelor’s degree or above, and 34.3% of those had a degree below a bachelor (see Figure 6).

Based on whether they have been to a beach or not, the proportion of respondents who have ever visited a beach is 62.4%; the proportion of respondents who have never visited a beach is only 37.6%. Those with beach visit experience were significantly better at judging the location of rips, with an accuracy of 61.7%. However, those without beach visit experience only exhibited an accuracy of 46.7%. The frequency of visiting the beach suggests that respondents who have never been to a beach accounted for 18.35%, and those who choose to visit the beach several times a month accounted for 40.64%.

Figure 7 indicates that the proportion of respondents who were unable to swim (50.6%) exceeded that of those who could swim (49.4%). However, there was no significant difference in the accuracy of judging the location of rips. The accuracy of choosing the escape direction was also similar, reflecting that whether they could swim had no impact on the judgment results. Swimming distance can represent swimming ability to some extent. In total, 57.5% of the respondents who have swimming experience in the sea do not exceed a swimming distance of 25 m, 27.5% of them have a swimming distance between 26–60 m, 7.5% have a swimming distance between 60–100 m, and 7.5% have a swimming distance over 100 m. Although those over 100 m were the most accurate in choosing the direction of escape, they were the worst at judging the location of rips.

Lifeguards are essential security on the beach for beachgoers. However, 11.5% of respondents had not seen lifeguards, 22.2% mentioned that there were lifeguards on only a few beaches, and 13.8% responded that they did not notice whether there were lifeguards or not. Only 27.8% had heard of the word “rip current”, and even fewer had seen a warning sign for rips (7.7%). Those who had heard about rips, though with a low proportion, exhibited a higher accuracy in judging the location of rips (61.6%) and choosing the direction of escape (40.4%) compared to those who had not heard about rips (47.7% and 36.3%). Respondents who had seen the warning signs were more accurate at judging the location of rips (50.5%) and choosing the direction of escape (46.7%) than those who had not seen the warning signs (40.4% and 35.9%).

In the field questionnaire, there were 168 male respondents, accounting for 54.9%, with an average age of 32.12; there were 138 females, accounting for 45.1%, with an average age of 31.97. Those with a bachelor’s degree or above accounted for 39.2% of the total; those with a degree below a bachelor’s accounted for 60.8% (see Figure 7).

The item “whether have they visited the beach” was omitted in the field survey. According to the data in Figure 6, those who visit the beach accounted for the highest proportion (34.3%); those who visit several times a week (29.4%) and those who visit several times a year (27.5%) are close to each other; those who visit every day only account for 5.9%. Regarding swimming ability, 57.8% of the beachgoers can swim, 42.2% cannot swim, and 58.8% have swum in the sea. This suggests that at least 1% of the beachgoers cannot swim but enter the water. If those who can swim but have not swum in the sea are excluded, the percentage is even higher. Particularly, the distance from the shore for the beachgoers who cannot swim was less than 20 m according to their statements, and the depth was less than 2 m. Meanwhile, most of them chose to wear swimming rings. Future investigations will examine this since it could be a crucial feature in lowering drowning accidents.

The proportion of respondents in the field choosing “heard of rips” and “having seen warning signs of rips” (35.9% and 20.6%) was higher than that of respondents in the network (27.8% and 7.7%). Nevertheless, the proportion was still extremely low. Concerning the question of whether there were lifeguards at the beaches they visited, 32.3% of the respondents answered that there were lifeguards at most beaches, 20.6% responded that there were lifeguards at a small number of beaches, 26.5% did not pay attention to them, 9.8% believed that there were lifeguards at all beaches, and 10.8% did not (see Figure 7). Overall, the accuracy of identifying the location of rips is close to the online survey (56.5% and 59.2%), and the accuracy of choosing the escape direction is also similar (39.2% and 37.8%).

The chi-square test results revealed that the visiting frequency ($C_{2}$) had a significant impact on the identification of rip currents ($\chi^{2}=16.96,\;p<0.05$) and the choice of escape direction ($\chi^{2}=28.09,\;p<0.05$). Variable $D_{1}$ had a significant impact on the identification of rip current ($\chi^{2}=14.02$, p<0.05) but not on the choice of escape direction. In addition, variable $D_{2}$ had significant impacts on both the recognition of rips and the choice of escape direction (see Table 4). In our results, swimming ability ($B_{3}$) might not be a determinant of rip identification, but a decisive factor in escape route selection ($\chi^{2}=15.74$, p<0.05).

### 3.3. Educational Strategies

In our study, the results of the two survey methods were compared to verify the implementation effect of the educational strategy. On the whole, there was very little difference between online and field respondents in the accuracy of identifying rips and choosing the correct escape route. However, the accuracy of the educated respondents in identifying the location of rip current increased from 56.5% to 90.5%, and the accuracy in choosing the escape direction increased from 39.2% to 85.9% (see Figure 8). Concerning demographic characteristics, respondents with higher education had higher accuracy in selecting the location of rip current, but this trend was not obvious when choosing the escape route. The results of the online survey also reflected this trend in general. Regarding swimming ability, beach visit frequency, and awareness of rips, the accuracy in choosing escape direction was also significantly improved (see Figure 9). This implied that our survey results had universal significance, and the educational strategy implemented was effective.

## 4. Discussion

To the best of our knowledge, this study is the first attempt to quantitatively evaluate the knowledge and judgment of beachgoers about rips in southern China. In this paper, the results are discussed in the context of existing literature.

### 4.1. Survey

Our survey results suggested that the perception of rips significantly varied among beachgoers of different age groups. The youth group (19–35 years old) had the highest proportion of “having heard about rips” (31.4%); the middle-aged group (36–59 years old) had the highest accuracy in choosing the escape direction (65.1%). The chi-square test demonstrated a significant relationship between age and the accuracy of choosing the right escape direction ($\chi^{2}=36.12,\;p<0.05$). The investigation results of Houser et al. [37] also verified a large possibility that young respondents were more likely to use the rip zone as a safe location. Moreover, no one in the children group (≤11 years old) had ever heard of “rips” or seen relevant warning signs. The reason why children are included in the investigation is that many of the beach drowning victims are children. In the year 2020 alone, more than 120 child drowning deaths were reported in mainland China [38].

Regarding gender, some foreign studies revealed that females were less likely to be involved in rip current events [39,40]. Morgan et al. [41] and Williamson et al. [42] believed that males were more likely than women to ignore danger warnings during surfing activities. However, Fallon et al. [23] discovered that male visitors to Miami Beach were well aware that rips were a major cause of drowning. Sherker et al. [20] revealed that age was not a significant factor influencing beachgoers’ swimming between beaches or in rips. Barlas and Beji [40] and Williamson et al. [42] pointed out that foreign beachgoers were less knowledgeable about beach safety than native beachgoers. They suggested that female beachgoers were more aware of rips. Nonetheless, there was no significant gender difference in the number of casualties caused by rips. According to our survey results, gender may not be an essential factor influencing the correct identification of the rip current. This is because there was little difference between males and females in the accuracy of identifying the rip current location (53.6% and 55%).

As indicated by the analysis of the survey results, after the intervention, the number of respondents choosing the correct escape direction accounted for 85.9% (i.e., 52.8% for males and 33.1% for females) (see Figure 8), and the number of respondents choosing wrongly or not knowing how to choose accounted for 14.1%. The number of males selecting the correct escape direction was 19.7% higher than that of females, while males constituted only 9.8% more participants than females in number, which demonstrated that males might outperform females in terms of escape. In the future, perhaps more female participants should be included when implementing educational strategies.

Although swimming ability is crucial for swimmers, the proportion of respondents who can swim was not optimistic in our results (49.4% in the online survey, 57.8% in the field survey). The statistical conclusions of Barlas and Beji [40], Paxton and Collins [43], and Woodward et al. [10] indicated that swimming ability and surf frequency were strongly correlated with the knowledge of rips. Drozdzewski et al. [44] and Williamson et al. [41] revealed that respondents with better swimming abilities had more knowledge of rips. However, Houser et al. [37] believed that people with strong swimming abilities tended to be more confident in their ability to identify rips. For those who swim in the water, it is easier to escape from a smaller flow velocity of rip, but difficult to escape from larger flow velocity rips [45]. In our results, respondents with the strongest self-rated swimming ability had the lowest accuracy in identifying the location of the rip current (16.7%), while their accuracy in choosing the escape direction was the highest (66.7%). This suggests that swimming ability affects the decision on which direction to escape in, but not the ability to identify the rip current. Perhaps overconfidence in one’s swimming ability accounts for the low recognition accuracy, which is consistent with the finding of Houser et al. [37].

The proportion of those who have heard of rips accounted for 27.8% and 35.9% of the respondents on the online survey and field survey, respectively; the proportion of those who have seen the warning signs of rips was lower (7.7% and 20.6%), reflecting the lack of intuitive awareness of rips and their risks among most beachgoers in China. 

### 4.2. Educational Strategies

To determine if our educational strategy affected participants’ awareness of rip hazard, we assessed beachgoers’ abilities to recognize rips and select the appropriate escape routes before and after the intervention. There are few beach safety presentations in China, despite the fact that they are available to beachgoers on numerous beaches in many different nations. Both in Australia and globally, there is a paucity of evidence-based research which has evaluated the effectiveness of this type of intervention [46,47,48]. Hatfield et al. [49] publicized knowledge of rips to the public through issuing posters, postcards, and brochures with the message of “Don’t get sucked in by the rip”, and achieved good intervention effects in the selection of swimming areas by beach visitors. Brander et al. [50] conducted a science-based beach safety presentation to increase public understanding of beach safety and awareness of hazards, confirming that direct presentations improved community understanding of beach safety practice and rip current awareness and identification, but might lead to overconfidence in the ability of identification. These methods are good educational strategies for public awareness of rips.

The findings of this study demonstrate that our instructional tactics had a significant positive effect. Participants were substantially more adept and assured at identifying rips when the intervention was put into practice. There was no hint of ignorance regarding rip currents among the respondents, and the responses to questions 13 and 14 were significantly more accurate. Numerous studies have shown that beachgoer ability to identify rips is poor [20,21,22,23,37,51], and all have recommended a need for increased education focusing on improving public recognition of rips.

A significant number of the participants in this study were willing to learn more about rips, as evidenced by their participation in the questionnaire. Following the intervention, the majority of participants correctly identified rips and made the appropriate escape route, which raised their desire to learn more about beach safety. After the intervention, however, some responders were still unable to identify rips and choose the correct escape route. We recorded respondents’ responses to their failure, including that beach states were not same, the visual difference between rips and general waves was not obvious, and there was a lack of adequate visual training. Thus, multiple interventions were recommended to enhance awareness of rips among beachgoers. In addition, different shapes and visually distinct rip images and videos were effective tools; for example, the use of images with colored dye release into rips helped to improve participants’ understanding of rips [52].

Menard et al. [53] point out the subjectivity involved in the choice of rip current image selection in terms of difficulty level of identifying the rip current. The precision of identification will vary depending on the distinct rips’ degrees of difficulty. As a result, we also advise using photographs with various challenges, such as images without rips, to intervene. In addition, attention was taken to select pictures that were taken from the viewpoint of being on the beach, as opposed to oblique images or satellite images. Uebelhoer et al. [54] used six images in a survey and found similar issues with levels of difficulty and ambiguity. The capacity of beachgoers to spot rips in practical situations may not be accurately reflected by the usage of photos with “obvious” rips. In a beach safety survey based on the Science of the Surf (SOS) presentation by Brander et al. [50], a varied qualitative selection of ‘difficulty’ was applied to the images used. Participants were asked to identify rips at different difficulty levels, thus reflecting their true ability to identify rips.

This study has some potential limitations. As noted by Menard et al. [53], there is a potential for false positives in survey-based rip identification studies which use static images, as the rip current is typically at, or near, the center of the photograph in order to properly show the associated wave-breaking pattern adjacent to the darker area of the channelized rip current. This is true for most of the images used in the surveys in this study (Figure A2 and Figure A3 in Appendix B). Moreover, static imagery also does not necessarily replicate real scenarios of rip conditions [46,49,53]. In light of this circumstance, we employed a video clip as part of our educational method, and rip-containing footage enables a more accurate and visual explanation of the rip current shape and properties. Other sorts of rips (hydrodynamically-controlled and boundary-controlled) are not shown in the video, which only displays one form of rip. Consequently, it is essential to integrate clips with various rip types in future instruction in order to improve educational outcomes. Regardless of the limitations of this study, the improvement in respondents’ ability to spot rips, awareness of rip identification characteristics, and expressed desire to learn more about beach safety demonstrate the potential and effectiveness of this type of intervention for all potential audiences.

## 5. Conclusions

In this study, beachgoers’ awareness of rips was investigated with online questionnaires and field questionnaires. The conclusions were drawn as follows.

The results of online and field questionnaires demonstrated that Chinese beachgoers were deficient in the awareness of rip current hazards, and most beachgoers had low safety awareness. The level of public awareness of rips affected their identification of rips and their choice of escape direction when encountering rips. We verified these findings through a regression model. An intervention with respondents using images and videos containing rips was used in the field survey, and significantly improved the accuracy of respondents in identifying rips and choosing escape directions. It is recommended that multiple interventions were recommended to enhance awareness of rips among beachgoers, especially among female participants where the intervention is more required. In future research, we will improve the questionnaire (such as by investigating some important factors of beach safety (e.g., attention to lifeguards and life-saving equipment, awareness of beach hazards before visiting, etc.), increasing the number of beach images or cards containing different types of rips, presenting the questionnaire using different languages), expand the survey scope, conduct more detailed questionnaire research on beaches with different states, and different shapes and visually distinct rips images will be used for educational intervention among beachgoers.

## Figures and Tables

**Figure 1 ijerph-20-04471-f001:**
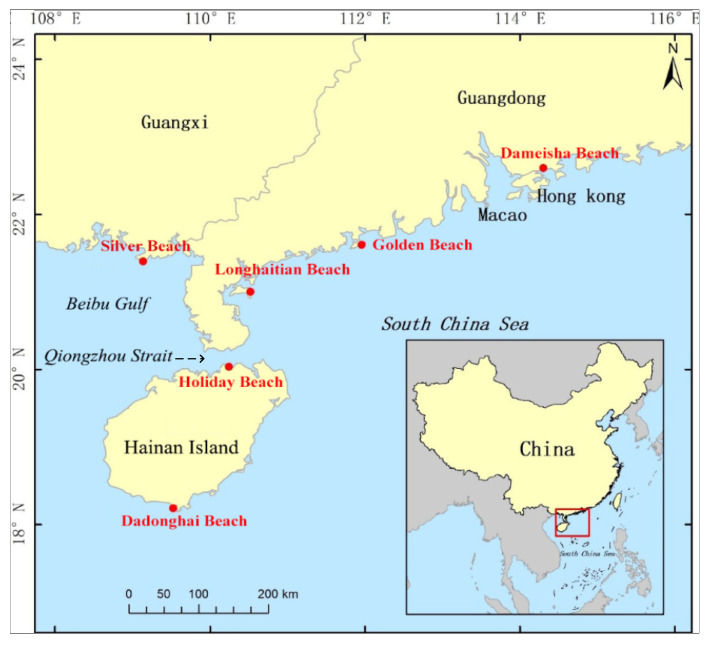
Location of the six beaches on the map of China. They are all famous tourist attractions in South China, namely, Silver Beach in Beihai City, Longhaitian Beach in Zhanjiang City, Hailing Island Gold Beach in Yangjiang City, Dameisha Beach in Shenzhen City, Holiday Beach in Haikou City, and Dadonghai Beach in Sanya City.

**Figure 2 ijerph-20-04471-f002:**
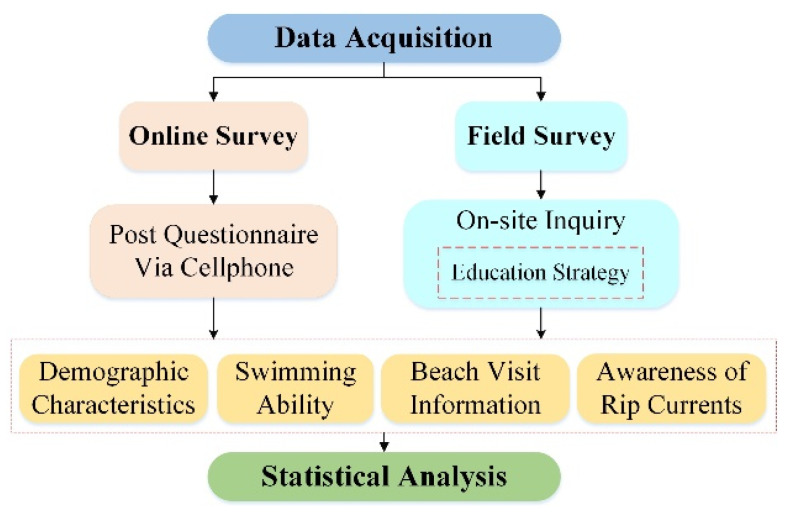
Flow chart of the survey methods.

**Figure 3 ijerph-20-04471-f003:**
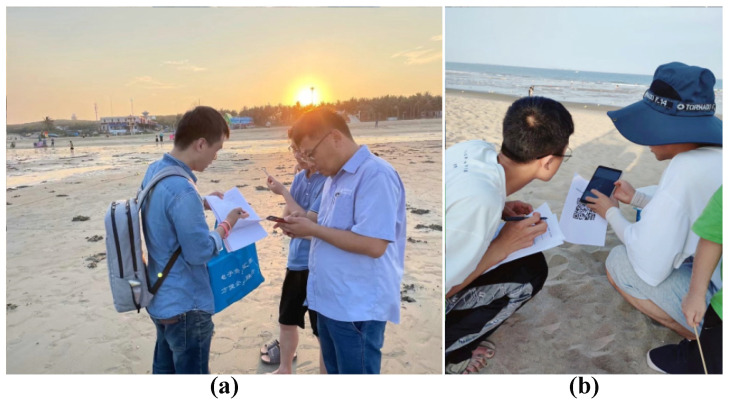
Flow chart of the survey methods. (**a**) Describe the structure and hazards of rips and the means of escape if trapped in a rip current; (**b**) display video showing the phenomenon of rips.

**Figure 4 ijerph-20-04471-f004:**
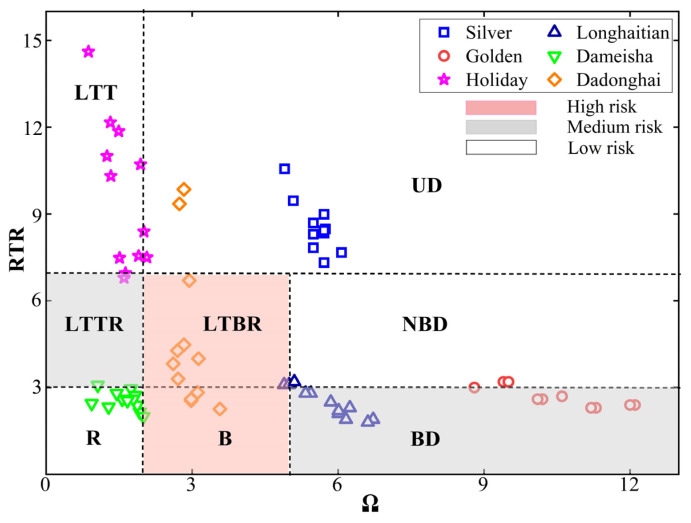
Monthly rip current hazards rating for six beaches based on morphodynamic analysis.

**Figure 5 ijerph-20-04471-f005:**
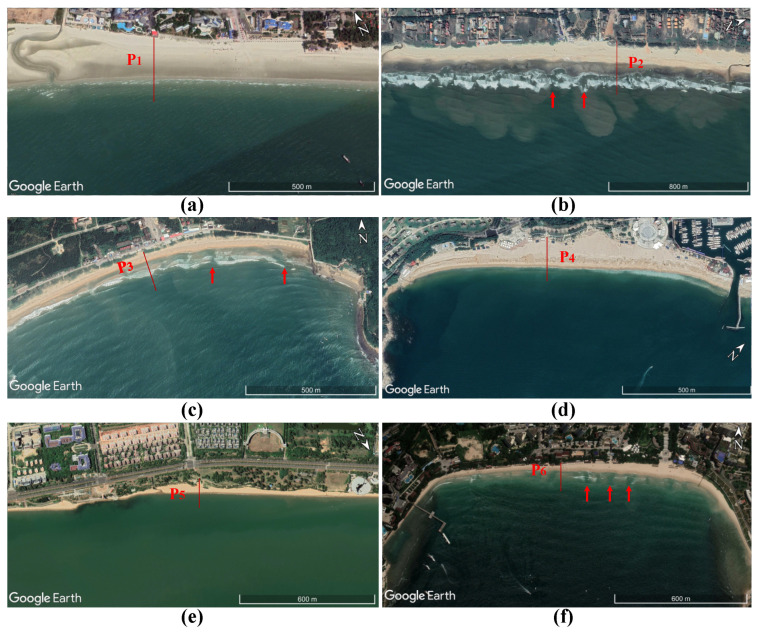
Google Earth image map of six beaches. (**a**) Silver Beach on 12 October 2019, (**b**) Longhaitian Beach on 3 November 2019, (**c**) Hailing Island Golden Beach on 7 September 2019, (**d**) Dameisha Beach on 5 October 2018, (**e**) Holiday Beach on 3 November 2019, (**f**) Dadonghai Beach on 4 August 2018. The red arrows indicate the location of rips, and the red lines indicate the location of profiles.

**Figure 6 ijerph-20-04471-f006:**
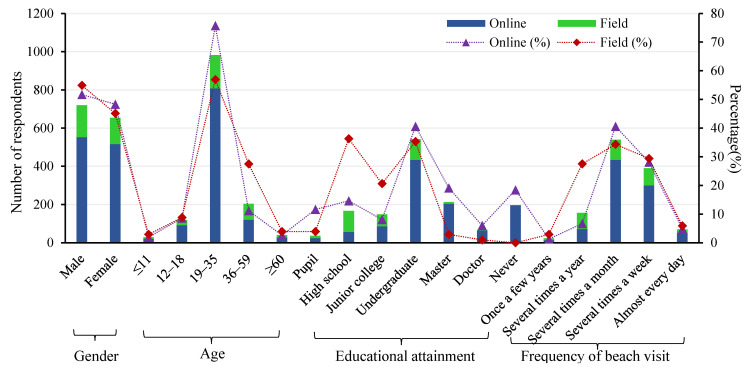
Demographic characteristics of online and field respondents, including gender, age, educational attainment and frequency of beach visit. The bar chart indicates the number of respondents, and the line chart indicates the percentage of each indicator.

**Figure 7 ijerph-20-04471-f007:**
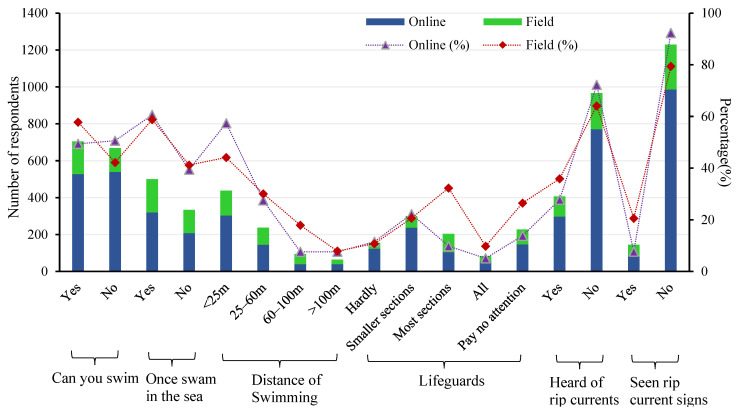
Respondents’ swimming ability and awareness of rips, and the distribution of beach lifeguards from the respondents’ responses. The bar chart indicates the number of respondents, and the line chart indicates the percentage of each indicator.

**Figure 8 ijerph-20-04471-f008:**
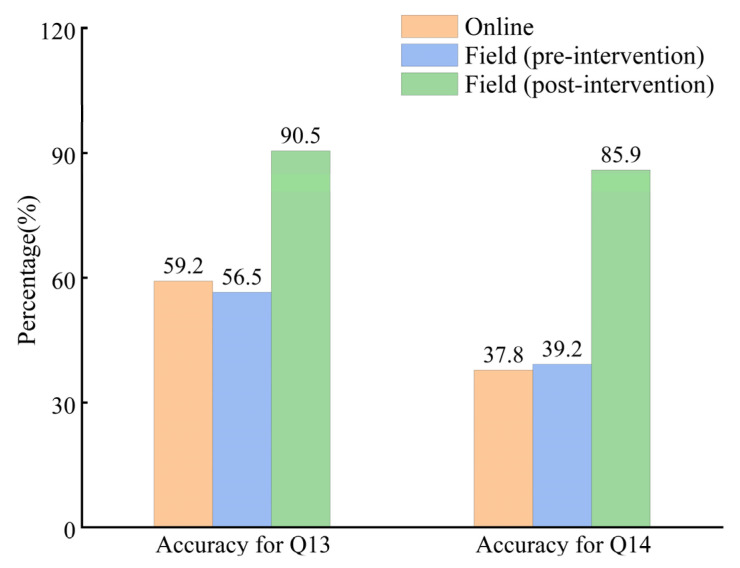
Comparison of the overall accuracy of online and field respondents in identifying rip current and choosing escape direction, as well as the accuracy after implementing educational strategy.

**Figure 9 ijerph-20-04471-f009:**
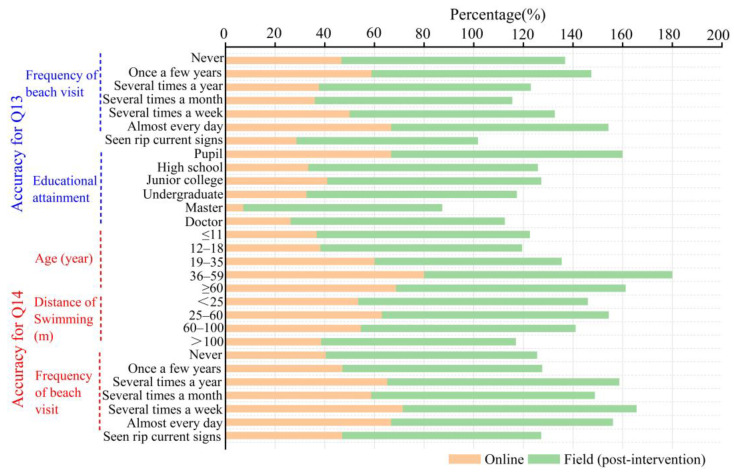
Influence of age, educational attainment, frequency of beach visit, swimming ability, and whether or not they have seen warning signs of rips on network and field respondents’ identification of rip current on actual beach scene and escape direction selection.

**Table 1 ijerph-20-04471-t001:** Visitor volume and hydrological conditions of beaches. *N* stands for yearly visitor reception, $H_{s}$ for the average significant wave height, *T* for the average significant wave period, *TR* for annual average spring tidal range, and $D_{50}$ for median sediment diameter.

Beach	*N* (Million)	$H_{s}$ (m)	*T* (s)	*TR* (m)	$D_{50}$ (mm)
Silver	2	0.20–0.30	2.20–2.50	3.50	0.22
Longhaitian	1.2	0.70–1.20	2.70–3.60	2.16	0.36
Gold	2	0.50–0.70	4.60–5.50	2.08	0.14
Dameisha	4	0.12–0.36	4.80–6.05	1.04	0.31
Holiday	1.2	0.20–0.50	3.40–4.90	2.19	0.39
Dadonghai	1.8	0.20–0.80	2.10–4.20	2.01	0.36

**Table 2 ijerph-20-04471-t002:** Beach state classification based on *RTR* and *Ω* parameters.

Beach Type	Ω	*RTR*	Rip Current Hazard
Complete reflective (R)	Ω<2	RTR<3	Low
Low tide terrace (LTT)	Ω<2	RTR>7	Low
Low tide terrace with rip (LTTR)	Ω<2	3≤RTR≤7	Medium
Barred (B)	2≤Ω≤5	RTR<3	High
Low tide bar/rip (LTBR)	2≤Ω≤5	3≤RTR≤7	High
Barred dissipative (BD)	Ω>5	RTR<3	Medium
Non-barred dissipative (NBD)	Ω>5	3≤RTR≤7	Low
Ultra-dissipative (UD)	Ω>2	RTR>7	Low

**Table 3 ijerph-20-04471-t003:** Variable types and definition descriptions in the model.

Types		Variables	Definitions
Independent variable (*X*)	Demographic characteristics (A)	Gender ($A_{1})$	Male = 1, Female = 2.
		Age ($A_{2}$)	≤11 year = 1, 12–18 year = 2, 19–35 year = 3, 36–59 year = 4, ≥60 year = 5.
		Education attainment ($A_{3}$)	Pupil = 1, High school = 2, Junior college = 3, Undergraduate = 4, Master = 5, Doctor = 6.
	Swimming ability (B)	Can you swim ($B_{1}$)	Y = 1, N = 0.
		Swim in the sea ($B_{2}$)	Y = 1, N = 0.
		Distance from the shore ($B_{3}$)	<25 m = 1, 25–60 m = 2, 60–100 m = 3, >100 m = 4.
	Beach visit information (C)	Visited a beach ($C_{1}$)	Y = 1, N = 0.
		Frequency ($C_{2}$)	Never = 1, Once a few years = 2, Several times a year = 3, Several times a month = 4, Several times a week = 5, Almost every day = 6.
		Lifeguards ($C_{3}$)	Pay no attention = 0, Hardly = 1, Small sections = 2, Most sections = 3, All = 4.
	Awareness of rip currents (D)	Heard of rips ($D_{1}$)	Y = 1, N = 0.
		Rip current warning signs ($D_{2}$)	Y = 1, N = 0.
Dependent variable (*Y*)	Identification	Identify the location of rips ($Y_{1}$)	Y = 1, N = 0.
		Choose the right escape route ($Y_{2}$)	Y = 1, N = 0.

**Table 4 ijerph-20-04471-t004:** The chi-square value between the independent and dependent variables; $P_{i}$ is the probability value calculated by the regression model.

$\mathrm{Independent}\;\mathrm{Variable}\;(X_{i})$	$\mathrm{Identification}\;\mathrm{of}\;\mathrm{Rips}\;(Y_{1})$			$\mathrm{Choose}\;\mathrm{the}\;\mathrm{Right}\;\mathrm{Escape}\;\mathrm{Route}\;(Y_{2})$		
	$\chi^{2}$	*p*	$P_{i}$	$\chi^{2}$	*p*	$P_{i}$
Gender ($A_{1}$)	0.824	0.364	0.161	14.582	0.243	0.158
Age ($A_{2}$)	6.315	0.177	0.250	36.123	0.032	0.261
Education attainment ($A_{3}$)	29.608	0.081	0.722	17.957	0.103	0.755
Can you swim ($B_{1})$	0.417	0.519	0.648	0.025	0.875	0.686
Swim in the sea ($B_{2}$)	0.531	0.466	0.490	0.021	0.885	0.522
Swimming distance ($B_{3}$)	9.142	0.418	0.859	15.746	0.005	0.466
Visited a beach ($C_{1}$)	1.551	0.213	0.764	7.355	0.093	0.674
Frequency ($C_{2}$)	16.967	0.005	0.031	28.092	0.005	0.096
Lifeguards ($C_{3})$	29.544	0.108	0.128	8.381	0.079	0.826
Heard of rips ($D_{1}$)	14.023	0.001	0.010	1.776	0.183	0.405
Rip warning signs ($D_{2}$)	4.712	0.030	0.036	4.491	0.034	0.038

## Data Availability

The data presented in this study are available on request from the corresponding author.

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
