# Peer review of "Preliminary Investigation and Analysis of Beachgoers’ Awareness of Rip Currents in South China"

_ijerph, 2023, doi:10.3390/ijerph20054471_

Round 1
Reviewer 1 Report
In this study, the authors tried to capture the beachgoers’ awareness of rip currents through demographic characteristics, swimming ability, information about the beaches, and knowledge about the rip currents. They observed that the beachgoers’ knowledge about the rip currents hazards is small. Also, they highlight the importance of educational strategies to increase knowledge about rip currents.
The structure of the paper could be clearer. I suggest:
1. Introduction, 2. Material and Methods (2.1 Research Area; 2.2. Morphodynamic beach classification; 2.3. Survey Methods; 2.4. Educational strategy; 2.5. Statistical analyses); 3. Results (3.1. Morphodynamic beach classification; 3.2. Survey; 3.3. Educational strategy); 4. Discussion( 4.1. Survey; 4.2 Educational strategies); 5. Conclusion.
The introduction has a good structure, with essential information about the hazards caused by rip currents around the world and the lack of knowledge by the beachgoers about the hazard. However, this section could be improved. The authors should explain how rip currents occur, the different types, and why they are so dangerous to beachgoers.
The paper of Castelle et al. 2016 reviewed the rip current types, circulation, and hazards, and it is a good reference to increase the introduction (http://dx.doi.org/10.1016/j.earscirev.2016.09.008).
Also, The third paragraph has much important information. However, as it is a long paragraph and most sentences start with the reference, the reading becomes exhausting. Therefore, I suggest inverting some sentences, starting with the ideas and the reference at the end.
The research area section is tiny, with low information about the six beaches. For me, just one table with hydrological and visitor volume is not enough. Please increase the description of the six beaches, for example, with information about morphodynamics and wave energy.
The beaches’ names in figure 1 are small, so it is impossible to read them. Increase the names on the map.
In the results, the authors should include a table with the results, including the Chi-square, and a table with the linear regression results. And, the discussion should be improved, mainly because most are results.
There needed to be more clarity in the sections. Some paragraphs are located in the wrong sections, for example:
- The first paragraph of the research area is not about the research area. Thus, I suggest splitting it. The first sentence will go to the introduction, “This is the first public survey of beachgoers at popular beaches in South China, combining online survey with field survey.”. In contrast, the rest will go to the Methods and Model section.
- The authors described in the 3.1 section how they calculated the rip current hazard and wrote them in the table in the section research area. This information is the results and should be in the results section. Thus, a new sub-topic in the results is necessary, “Morphodynamic beach classification.” And the information about the Rip current hazard, figure 2, and the text between lines 146-152 should be included in this new sub-topic in the results.
- In the discussion section, the results about the educational strategy and model validation should be in the results.
Some specifics comments:
Line 171- The six beaches described in Section 2.
Lines 396-398: should be in the methods section (except Figure 9).
Line 44-45: References in a different format.
Reviewer 2 Report
I have carefully read the paper and I believe that it needs a revision concerning its language. It was very difficult to read some of the sections of the manuscript. Also, some revision is needed regarding the structure and the references of the manuscript. On the other hand, the study is interesting and may consider publication after major revision. The authors through their paper they conducted a public survey (online and field questionnaire) at popular beaches in South China concerning knowledge about rip current hazards to the public. Also, through their research they demonstrated the benefits of the application of a wide educational strategy about rip currents.
General comments:
1. The manuscript in general follows the MDPI template. Some corrections regarding the references must be addressed.
2. The use of the English language needs a detailed revision.
3. Some methodological approaches must be explained in detail.
For specific comments, please refer in the manuscript.
Overall, my suggestion is that the paper should be consider for publishing after major revision.

Round 2
Reviewer 1 Report
I would like to thank the authors for addressing my comments. However, there are a few things in the document that should be addressed.

Author Response
Thank you for your comments and valuable suggestions about the manuscript, the revisions have been submitted via the link. Please see the attachment.
